# Neuroscout, a unified platform for generalizable and reproducible fMRI research

**Alejandro de la Vega**[1][\*][†], **Roberta Rocca**[1,2][†], **Ross W Blair**[3],
**Christopher J Markiewicz**[3], **Jeff Mentch**[4,5], **James D Kent**[1], **Peer Herholz**[6],
**Satrajit S Ghosh**[5,7], **Russell A Poldrack**[3], **Tal Yarkoni**[1]

[1]Department of Psychology, The University of Texas at Austin, Austin, United States; [2]Interacting Minds Centre, Aarhus University, Aarhus, Denmark; [3]Department of Psychology, Stanford University, Stanford, United States; [4]Program in Speech and Hearing Bioscience and Technology, Harvard University, Cambridge, United States; [5]McGovern Institute for Brain Research, Massachusetts Institute of Technology, Cambridge, United States; [6]McConnell Brain Imaging Centre, Montreal Neurological Institute, McGill University, Montreal, Canada; [7]Department of Otolaryngology, Harvard Medical School, Boston, United States

**Abstract** Functional magnetic resonance imaging (fMRI) has revolutionized cognitive neuroscience, but methodological barriers limit the generalizability of findings from the lab to the real world. Here, we present Neuroscout, an end-to-end platform for analysis of naturalistic fMRI data designed to facilitate the adoption of robust and generalizable research practices. Neuroscout leverages state-of-the-art machine learning models to automatically annotate stimuli from dozens of fMRI studies using naturalistic stimuli—such as movies and narratives—allowing researchers to easily test neuroscientific hypotheses across multiple ecologically-valid datasets. In addition, Neuroscout builds on a robust ecosystem of open tools and standards to provide an easy-to-use analysis builder and a fully automated execution engine that reduce the burden of reproducible research. Through a series of meta-analytic case studies, we validate the automatic feature extraction approach and demonstrate its potential to support more robust fMRI research. Owing to its ease of use and a high degree of automation, Neuroscout makes it possible to overcome modeling challenges commonly arising in naturalistic analysis and to easily scale analyses within and across datasets, democratizing generalizable fMRI research.

**\*For correspondence:**
delavega@utexas.edu

[†]These authors contributed equally to this work

**Competing interest:** The authors declare that no competing interests exist.

## Editor's evaluation

This is an important, methodologically compelling paper. It describes a powerful new online software platform for analysing data from naturalistic fMRI studies. The paper describes both the philosophy behind and intended usage of the software, and offers several examples of the types of results that can be computed using publicly available datasets. It will provide an important new tool for the open neuroscience community who are seeking to perform standardised and reproducible analyses of naturalistic fMRI datasets.

## Introduction

Functional magnetic resonance imaging (fMRI) is a popular tool for investigating how the brain supports real-world cognition and behavior. Vast amounts of resources have been invested in fMRI

research, and thousands of fMRI studies mapping cognitive functions to brain anatomy are published every year. Yet, increasingly urgent methodological concerns threaten the reliability of fMRI results and their generalizability from laboratory conditions to the real world.

A key weakness of current fMRI research concerns its generalizability—that is, whether conclusions drawn from individual studies apply beyond the participant sample and experimental conditions of the original study (*Turner et al., 2018*; *Bossier et al., 2020*; *Yarkoni, 2020*; *Szucs and Ioannidis, 2017*). A major concern is the type of stimuli used in the majority of fMRI research. Many studies attempt to isolate cognitive constructs using highly controlled and limited sets of reductive stimuli, such as still images depicting specific classes of objects in isolation, or pure tones. However, such stimuli radically differ in complexity and cognitive demand from real-world contexts, calling into question whether resulting inferences generalize outside the laboratory to more ecological settings (*Nastase et al., 2020*). In addition, predominant statistical analysis approaches generally fail to model stimulus-related variability. As a result, many studies–and especially those relying on small stimulus sets–likely overestimate the strength of their statistical evidence and their generalizability to new but equivalent stimuli (*Westfall et al., 2016*). Finally, since fMRI studies are frequently underpowered due to the cost of data collection, results can fail to replicate on new participant samples (*Button et al., 2013*; *Cremers et al., 2017*).

Naturalistic paradigms using life-like stimuli have been advocated as a way to increase the generalizability of fMRI studies (*DuPre et al., 2020*; *Hamilton and Huth, 2020*; *Nastase et al., 2020*; *Sonkusare et al., 2019*). Stimuli such as movies and narratives feature rich, multidimensional variation, presenting an opportunity to test hypotheses from highly controlled experiments in more ecological settings. Yet, despite the proliferation of openly available naturalistic datasets, challenges in modeling these data limit their impact. Naturalistic features are difficult to characterize and co-occur with potential confounds in complex and unexpected ways (*Nastase et al., 2020*). This is exacerbated by the laborious task of annotating events at fine temporal resolution, which limits the number of variables that can realistically be defined and modelled. As a result, isolating relationships between specific features of the stimuli and brain activity in naturalistic data is especially challenging, which deters researchers from conducting naturalistic experiments and limiting re-use of existing public datasets.

A related and more fundamental concern limiting the impact of fMRI research is the low reproducibility of analysis workflows. Incomplete reporting practices in combination with flexible and variable analysis methods (*Carp, 2012*) are a major culprit. For instance, a recent large-scale effort to test identical hypotheses in the same dataset by 70 teams found a high degree of variability in the results, with different teams often reaching different conclusions (*Botvinik-Nezer et al., 2020*). Even re-executing the original analysis from an existing publication is rarely possible, due to insufficient provenance and a reliance on exclusively verbal descriptions of statistical models and analytical workflows (*Ghosh et al., 2017*; *Mackenzie-Graham et al., 2008*).

The recent proliferation of community-led tools and standards—most notably the Brain Imaging Data Structure (*Gorgolewski et al., 2016*) standard—has galvanized efforts to foster reproducible practices across the data analysis lifecycle. A growing number of data archives, such as OpenNeuro (*Markiewicz et al., 2021c*), now host hundreds of publicly available neuroimaging datasets, including dozens of naturalistic fMRI datasets. The development of standardized quality control and preprocessing pipelines, such as MRIQC (*Esteban et al., 2017*), fmriprep (*Esteban et al., 2019*; *Esteban et al., 2022*), and C-PAC (*Craddock et al., 2013*), facilitate their analysis and can be launched on emerging cloud-based platforms, such as https://brainlife.io/about/ (*Avesani et al., 2019*). However, fMRI model specification and estimation remains challenging to standardize, and typically results in bespoke modeling pipelines that are not often shared, and can be difficult to re-use. Unfortunately, despite the availability of a rich ecosystem of tools, assembling them into a complete and reproducible workflow remains out of reach for many scientists due to substantial technical challenges.

In response to these challenges, we developed Neuroscout: a unified platform for generalizable and reproducible analysis of naturalistic fMRI data. Neuroscout improves current research practice in three key ways. First, Neuroscout provides an easy-to-use interface for reproducible analysis of BIDS datasets, seamlessly integrating a diverse ecosystem of community-developed resources into a unified workflow. Second, Neuroscout encourages re-analysis of public naturalistic datasets by providing access to hundreds of predictors extracted through an expandable set of state-of-the-art feature extraction algorithms spanning multiple stimulus modalities. Finally, by using standardized

model specifications and automated workflows, Neuroscout enables researchers to easily operation-alize hypotheses in a uniform way across multiple (and diverse) datasets, facilitating more generaliz-able multi-dataset workflows such as meta-analysis.

In the following, we provide a broad overview of the Neuroscout platform, and validate it by repli-cating well-established cognitive neuroscience findings using a diverse set of public naturalistic data-sets. In addition, we present two case studies—face sensitivity of the fusiform face area and selectivity to word frequency in visual word form area—to show how Neuroscout can be used to conduct original research on public naturalistic data. Through these examples, we demonstrate how Neuroscout's flex-ible interface and wide range of predictors make it possible to dynamically refine models and draw robust inference on naturalistic data, while simultaneously democratizing gold standard practices for reproducible research.

## Results

### Overview of the Neuroscout platform

At its core, Neuroscout is a platform for reproducible fMRI research, encompassing the complete lifecycle of fMRI analysis from model specification and estimation to the dissemination of results. We focus particular attention on encouraging the re-use of public datasets that use intrinsically high dimensional and generalizable naturalistic stimuli such as movies and audio narratives. The platform is composed of three primary components: a data ingestion and feature extraction server, interactive analysis creation tools, and an automated model fitting workflow. All elements of the platform are seamlessly integrated and can be accessed interactively online (https://neuroscout.org). Complete

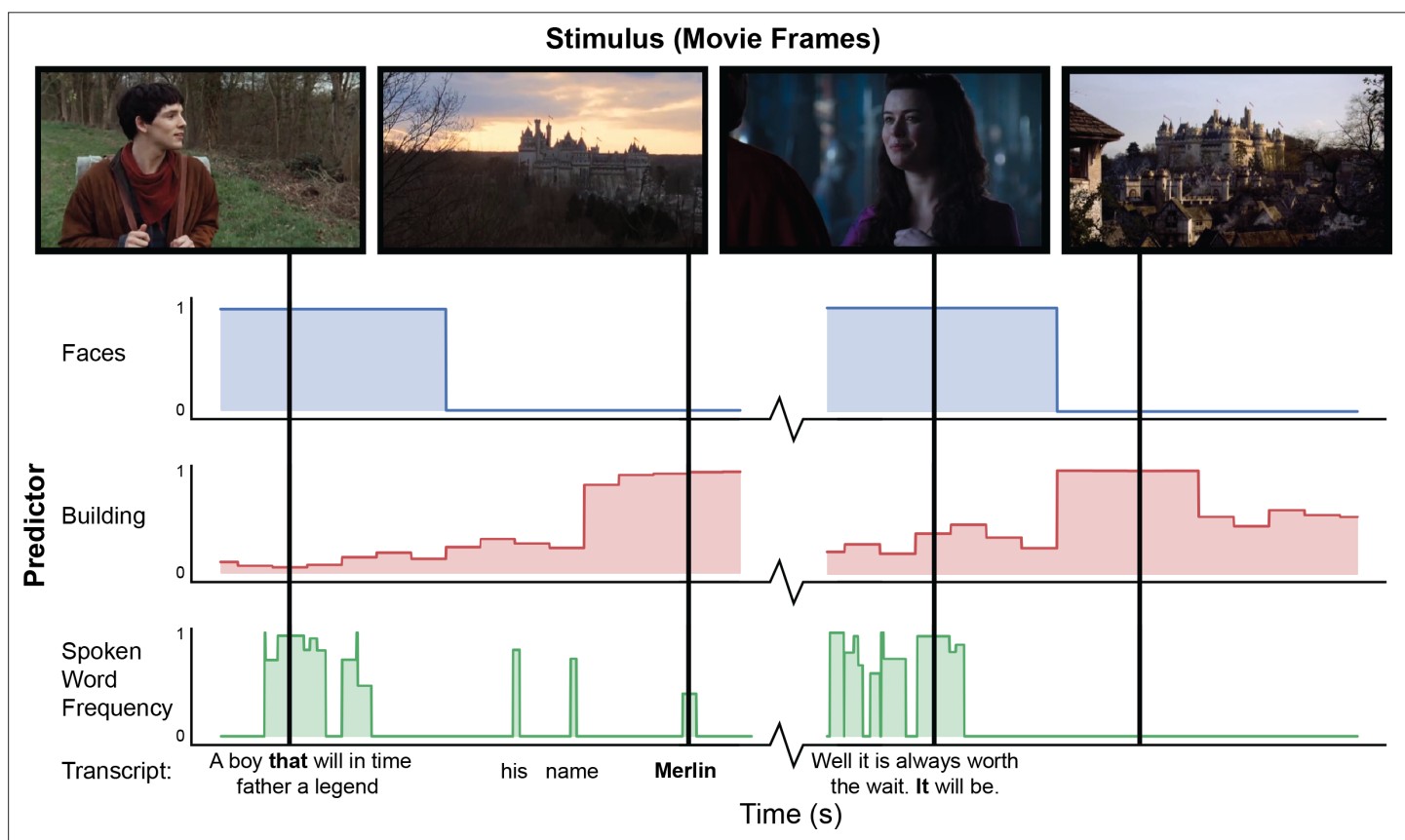

**Figure 1.** Example of automated feature extraction on stimuli from the "Merlin" dataset. Visual features were extracted from video stimuli at a frequency of 1 Hz. 'Faces': we applied a well-validated cascaded convolutional network trained to detect the presence of faces (*Zhang et al., 2016*). 'Building': We used Clarifai's General Image Recognition model to compute the probability of the presence of buildings in each frame. 'Spoken word frequency' codes for the lexical frequency of words in the transcript, as determined by the SubtlexUS database (*Brysbaert and New, 2009*). Language features are extracted using speech transcripts with precise word-by-word timing determined through forced alignment.

and up-to-date documentation of all of the platform's components, including Getting Started guides to facilitate first time users, is available in the official Neuroscout Documentation (https://neuroscout.org/docs).

## Preprocessed and harmonized naturalistic fMRI datasets

The Neuroscout server indexes a curated set of publicly available naturalistic fMRI datasets, and hosts automatically extracted annotations of visual, auditory, and linguistic events from the experimental stimuli. Datasets are harmonized, preprocessed, and ingested into a database using robust BIDS-compliant pipelines, facilitating future expansion.

## Automated annotation of stimuli

Annotations of stimuli are automatically extracted using *pliers* (*McNamara et al., 2017*), a comprehensive feature extraction framework supporting state-of-the-art algorithms and deep learning models (*Figure 1*). Currently available features include hundreds of predictors coding for both low-level (e.g. brightness, loudness) and mid-level (e.g. object recognition indicators) properties of audio-visual stimuli, as well as natural language properties from force aligned speech transcripts (e.g. lexical frequency annotations). The set of available predictors can be easily expanded through community-driven implementation of new *pliers* extractors, as well as publicly shared repositories of deep learning models, such as HuggingFace (*Wolf et al., 2020*) and TensorFlowHub (*Abadi et al., 2015*). We expect that as machine learning models continue to evolve, it will be possible automatically extract higher level features from naturalistic stimuli. All extracted predictors are made publicly available through a well-documented application programming interface (https://neuroscout.org/api). An interactive web tool that makes it possible to further refine extracted features through expert human curation is currently under development.

## Analysis creation and execution tools

Neuroscout's interactive analysis creation tools—available as a web application (https://neuroscout.org/builder) and python library (pyNS)—enable easy creation of fully reproducible fMRI analyses (*Figure 2a*). To build an analysis, users choose a dataset and task to analyze, select among pre-extracted

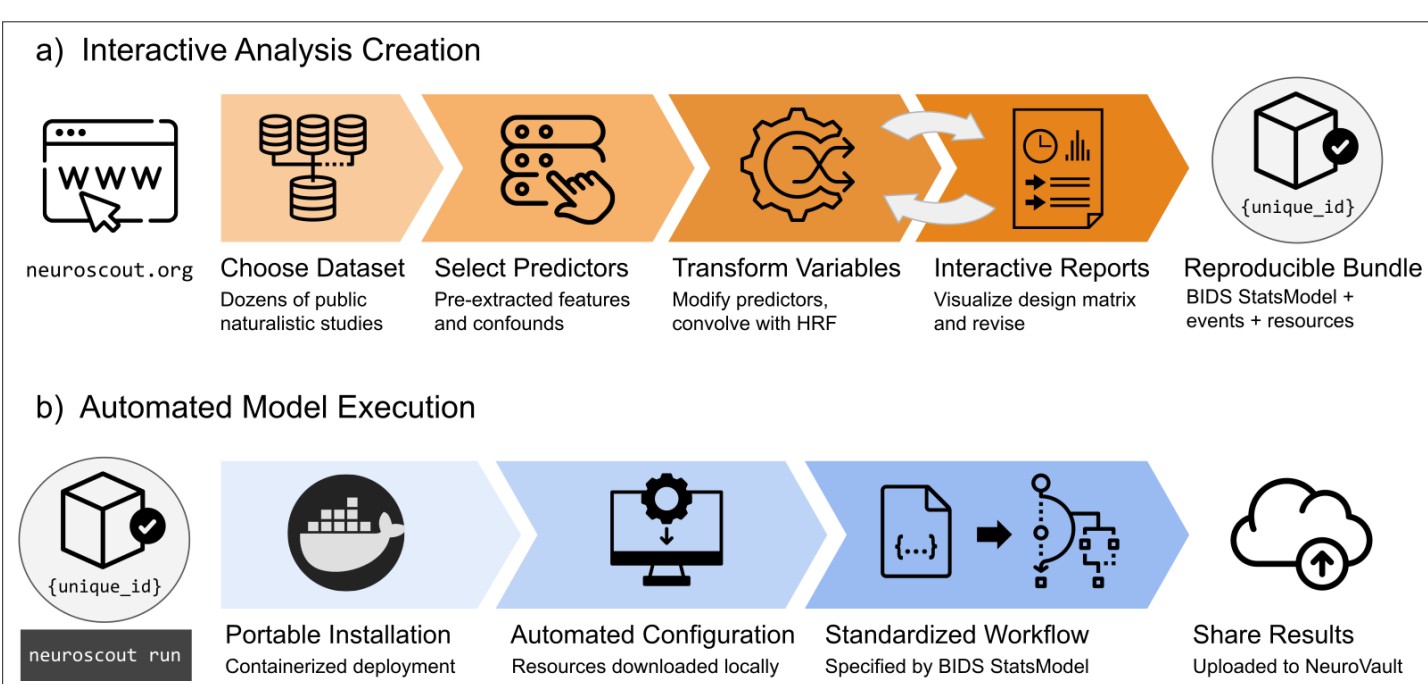

**Figure 2.** Overview schematic of analysis creation and model execution. (**a**) Interactive analysis creation is made possible through an easy-to-use web application, resulting in a fully specified reproducible analysis bundle. (**b**) Automated model execution is achieved with little-to-no configuration through a containerized model fitting workflow. Results are automatically made available in NeuroVault, a public repository for statistical maps.

predictors and nuisance confounds to include in the model, and specify statistical contrasts. Raw predictor values can be modified by applying model-specific variable transformations such as scaling, thresholding, orthogonalization, and hemodynamic convolution. Internally all elements of the multi-level statistical model are formally represented using the BIDS Statistical Models specification (*Markiewicz et al., 2021a*), ensuring transparency and reproducibility. At this point, users can inspect the model through quality-control reports and interactive visualizations of the design matrix and predictor covariance matrix, iteratively refining models if necessary. Finalized analyses are locked from further modification, assigned a unique identifier, and packaged into a self-contained bundle.

Analyses can be executed in a single command line using Neuroscout's automated model execution workflow (*Figure 2b*). Neuroscout uses container technology (i.e. Docker and Singularity) to minimize software dependencies, facilitate installation, and ensure portability across a wide range of environments (including high performance computers (HPC) and the cloud). At run time, preprocessed imaging data are automatically fetched using DataLad (*Halchenko et al., 2021*), and the analysis is executed using FitLins (*Markiewicz et al., 2021b*), a standardized pipeline for estimating BIDS Stats Models. Once completed, thresholded statistical maps and provenance metadata are submitted to NeuroVault (*Gorgolewski et al., 2015*), a public repository for statistical maps, guaranteeing compliance to FAIR (findable, accessible, interoperable, and reusable) scientific principles (*Wilkinson et al., 2016*). Finally, Neuroscout facilitates sharing and appropriately crediting the dataset and tools used in the analysis by automatically generating a bibliography that can be used in original research reports.

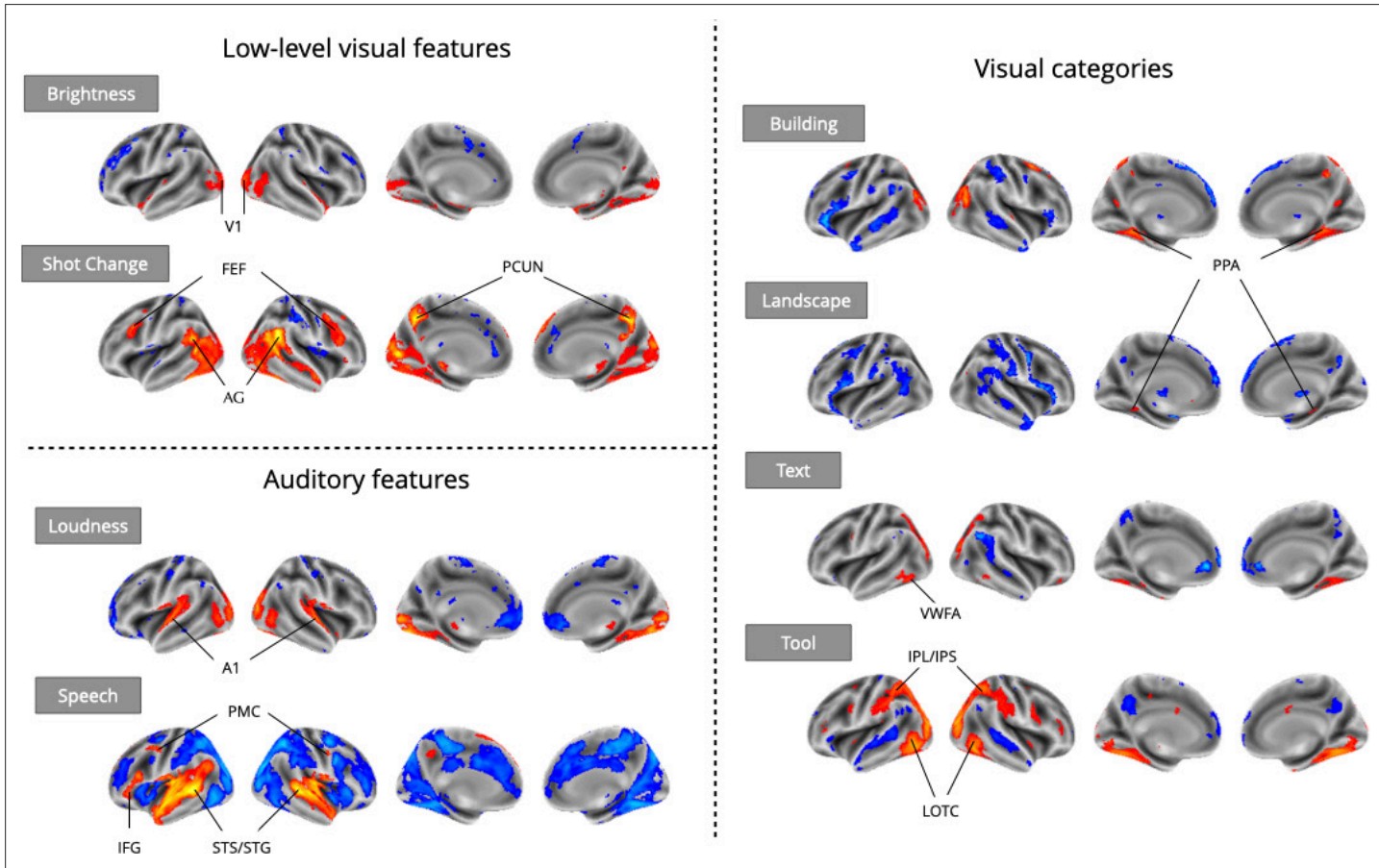

**Figure 3.** Meta-analytic statistical maps for GLM models targeting a variety of effects with strong priors from fMRI research. Individual GLM models were fit for each effect of interest, and dataset level estimates were combined using image-based meta-analysis. Images were thresholded at Z=3.29 (*P*<0.001) voxel-wise. Abbreviations: V1=primary visual cortex; FEF = frontal eye fields; AG = angular gyrus; PCUN = precuneus; A1=primary auditory cortex; PMC = premotor cortex; IFG = inferior frontal gyrus; STS = superior temporal sulcus; STG = superior temporal gyrus; PPA = parahippocampal place area; VWFA = visual word-form area; IPL = inferior parietal lobule; IPS = inferior parietal sulcus; LOTC = lateral occipito-temporal cortex.

## Scalable workflows for generalizable inference

Neuroscout makes it trivial to specify and analyze fMRI data in a way that meets gold standard reproducibility principles. This is per se a crucial contribution to fMRI research, which often fails basic reproducibility standards. However, Neuroscout's transformative potential is fully realized through the scalability of its workflows. Automated feature extraction and standardized model specification make it easy to operationalize and test equivalent hypotheses across many datasets, spanning larger participant samples and a more diverse range of stimuli.

The following analyses demonstrate the potential of multi-dataset approaches and their importance for generalizable inference by investigating a set of well-established fMRI findings across all of Neuroscout's datasets. We focused these analyses on three feature modalities (visual, auditory, and language), ranging from low-level features of the signal (loudness, brightness, presence of speech, and shot change), to mid-level characteristics with well established focal correlates (visual presence of buildings, faces, tools, landscape and text). For each feature and stimulus, we fit a whole-brain univariate GLM with the target feature as the sole predictor, in addition to standard nuisance covariates (see Methods for details). Finally, we combined estimates across twenty studies using random-effects image-based meta-analysis (IBMA), resulting in a consensus statistical map for each feature.

Even using a simple one-predictor approach, we observed robust meta-analytic activation patterns largely consistent with expectations from the existing literature (*Figure 3*), a strong sign of the reliability of automatically extracted predictors. We observed activation in the primary visual cortex for brightness (*Peters et al., 2010*), parahippocampal place area (PPA) activation in response to buildings and landscapes (*Park and Chun, 2009*; *Häusler et al., 2022*), visual word form area (VWFA) activation in response to text (*Chen et al., 2019*), and lateral occipito-temporal cortex (LOTC) and parietal activation in regions associated with action perception and action knowledge (*Schone et al., 2021*; *Valyear et al., 2007*) in response to the presence of tools on screen. For auditory features, we observed primary auditory cortex activation in response to loudness (*Langers et al., 2007*), and superior temporal sulcus and gyrus activity in response to speech (*Sekiyama et al., 2003*). We also observed plausible results for visual shot changes, a feature with fewer direct analogs from the literature, which yielded activations in the frontal eye fields, the precuneus, and parietal regions areas traditionally implicated in attentional orienting and reference frame shifts (*Corbetta et al., 1998*; *Fox et al., 2006*; *Kravitz et al., 2011*; *Rocca et al., 2020*). The only notable exception was a failure to detect fusiform face area (FFA) activity in response to faces (Figure 5), an interesting result that we dissect in the following section.

Crucially, although study-level results largely exhibited plausible activation patterns, a wide range of idiosyncratic variation was evident across datasets (*Figure 4*). For instance, for 'building' we observed PPA activity in almost every study. However, we observed a divergent pattern of activity in the anterior temporal lobe (ATL), with some studies indicating a deactivation, others activation, and others no relationship. This dissonance was resolved in the meta-analysis, which indicated no relationship with 'building' and the ATL, but confirmed a strong association with the PPA. Similar study-specific variation can be observed with other features. These results highlight the limits of inferences made from single datasets, which could lead to drawing overly general conclusions. In contrast, multi-dataset meta-analytic approaches are intrinsically more robust to stimulus-specific variation, licensing broader generalization.

## Flexible covariate addition for robust naturalistic analysis

A notable exception to the successful replications presented in the previous section is the absence of fusiform face area (FFA) activation for faces in naturalistic stimuli (*Figure 5*). Given long-standing prior evidence implicating the FFA in face processing (*Kanwisher et al., 1997*), it is highly unlikely that these results are indicative of flaws in the extant literature. A more plausible explanation is that our 'naive' single predictor models failed to account for complex scene dynamics present in naturalistic stimuli. Unlike controlled experimental designs, naturalistic stimuli are characterized by systematic co-occurrences between cognitively relevant events. For example, in narrative-driven movies (the most commonly used audio-visual naturalistic stimuli) the presentation of faces often co-occurs with speech—a strong driver of brain activity. Failing to account for this shared variance can confound model estimates and mask true effects attributable to predictors of interest.

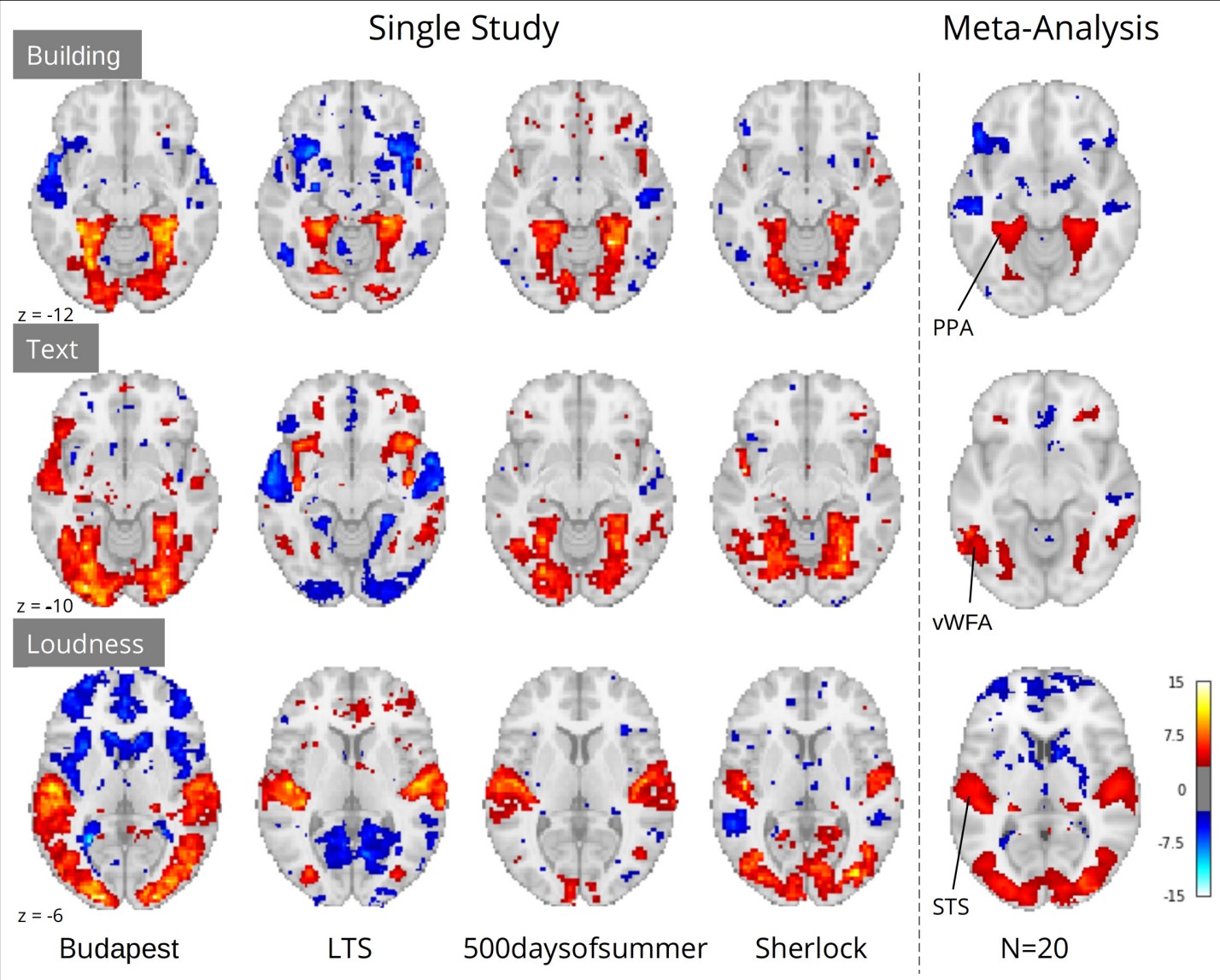

**Figure 4.** Comparison of a sample of four single study results with meta-analysis (N=20) for three features: 'building' and 'text' extracted through Clarifai visual scene detection models, and sound 'loudness' (root mean squared of the auditory signal). Images were thresholded at Z=3.29 (p<0.001) voxel-wise. Regions with a priori association with each predictor are highlighted: PPA, parahippocampal place area; VWFA, visual word form area; STS, superior temporal sulcus. Datasets: Budapest, Learning Temporal Structure (LTS), 500daysofsummer task from Naturalistic Neuroimaging Database, and Sherlock.

Neuroscout addresses these challenges by pairing access to a wide range of pre-extracted features with a flexible and scalable model specification framework. Researchers can use Neuroscout's model builder to iteratively build models that control and assess the impact of a wide range of potential confounds without the need for additional data collection or manual feature annotation. Analysis reports provide visualizations of the correlation structure of design matrices, which can inform covariate selection and facilitate interpretation. These affordances for iterative covariate control allow us to readily account for the potential confounding effect of speech, a predictor that co-varies with faces in some datasets but not others (Pearson's R range: –0.55, 0.57; mean: 0.18). After controlling for speech, we observed an association between face presentation and right FFA activity across 17 datasets (*Figure 5*; peak z=5.70). Yet, the strength of this relationship remained weaker than one might expect from traditional face localizer tasks.

In movies, face perception involves repeated and protracted presentation of a relatively narrow set of individual faces. Given evidence of rapid adaptation of category-selective fMRI response to

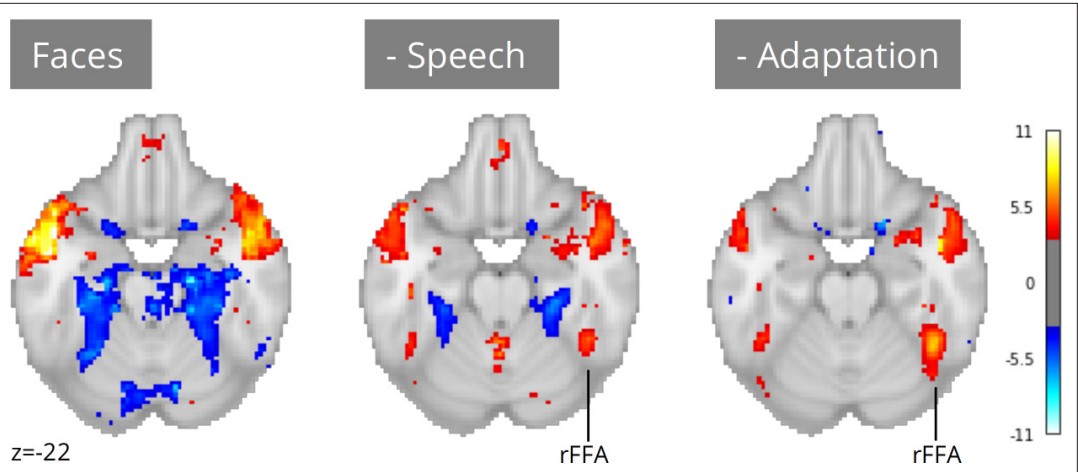

**Figure 5.** Meta-analysis of face perception with iterative addition of covariates. Left; Only including binary predictors coding for the presence of faces on screen did not reveal activity in the right fusiform face area (rFFA). Middle; Controlling for speech removed spurious activations and revealed rFFA association with face presentation. Right; Controlling for temporal adaptation to face identity in addition to speech further strengthened the association between rFFA and face presentation. N=17 datasets; images were thresholded at Z=3.29 (p<0.001) voxel-wise.

individual stimuli (*Grill-Spector et al., 1999*), FFA activation in naturalistic stimuli may be attenuated by a failure to distinguish transient processes (e.g. initial encoding) from indiscriminate face exposure. To test the hypothesis that adaptation to specific faces suppresses FFA activity, we further refined our models by controlling for the cumulative time of exposure to face identities (in addition to controlling for speech). Using embeddings from FaceNet, a face recognition convolutional neural network, we clustered individual face presentations into groups representing distinct characters in each movie. We then computed the cumulative presentation of each face identity and included this regressor as a covariate.

After controlling for face adaptation, we observed stronger effects in the right FFA (*Figure 5*; peak z=7.35), highlighting its sensitivity to dynamic characteristics of face presentation which cannot always be captured by traditional designs. Notably, unlike in traditional localizer tasks, we still observe significant activation outside of the FFA, areas whose relation to face perception can be further explored in future analyses using Neuroscout's rich feature set.

## Large samples meet diverse stimuli: a linguistic case study

Our final example illustrates the importance of workflow scalability in the domain of language processing, where the use of naturalistic input has been explicitly identified as not only beneficial but necessary for real-world generalizability (*Hamilton and Huth, 2020*). Owing to their ability to provide more robust insights into real-life language processing, studies using naturalistic input (e.g. long written texts or narratives) are becoming increasingly common in language neuroscience (*Andric and Small, 2015*; *Brennan, 2016*; *Nastase et al., 2021*). Yet, even when naturalistic stimuli are used, individual studies are rarely representative of the many contexts in which language production and comprehension take place in daily life (e.g. dialogues, narratives, written exchanges, etc), which raises concerns on the generalizability of their findings. Additionally, modeling covariates is particularly challenging for linguistic stimuli, due to their complex hierarchical structure. As a consequence, single studies are often at risk of lacking the power required to disentangle the independent contributions of multiple variables.

A concrete example of this scenario comes from one of the authors' (TY) previous work (*Yarkoni et al., 2008*). In a naturalistic rapid serial visual presentation (RSVP) reading experiment, *Yarkoni et al., 2008* reported an interesting incidental result: activity in the visual word form area (VWFA)—an area primarily associated with visual feature detection and orthography-phonology mapping (*Dietz et al., 2005*)—was significantly modulated by lexical frequency. Interestingly, these effects were robust to phonological and orthographic covariates, suggesting that VWFA activity may not only be involved in

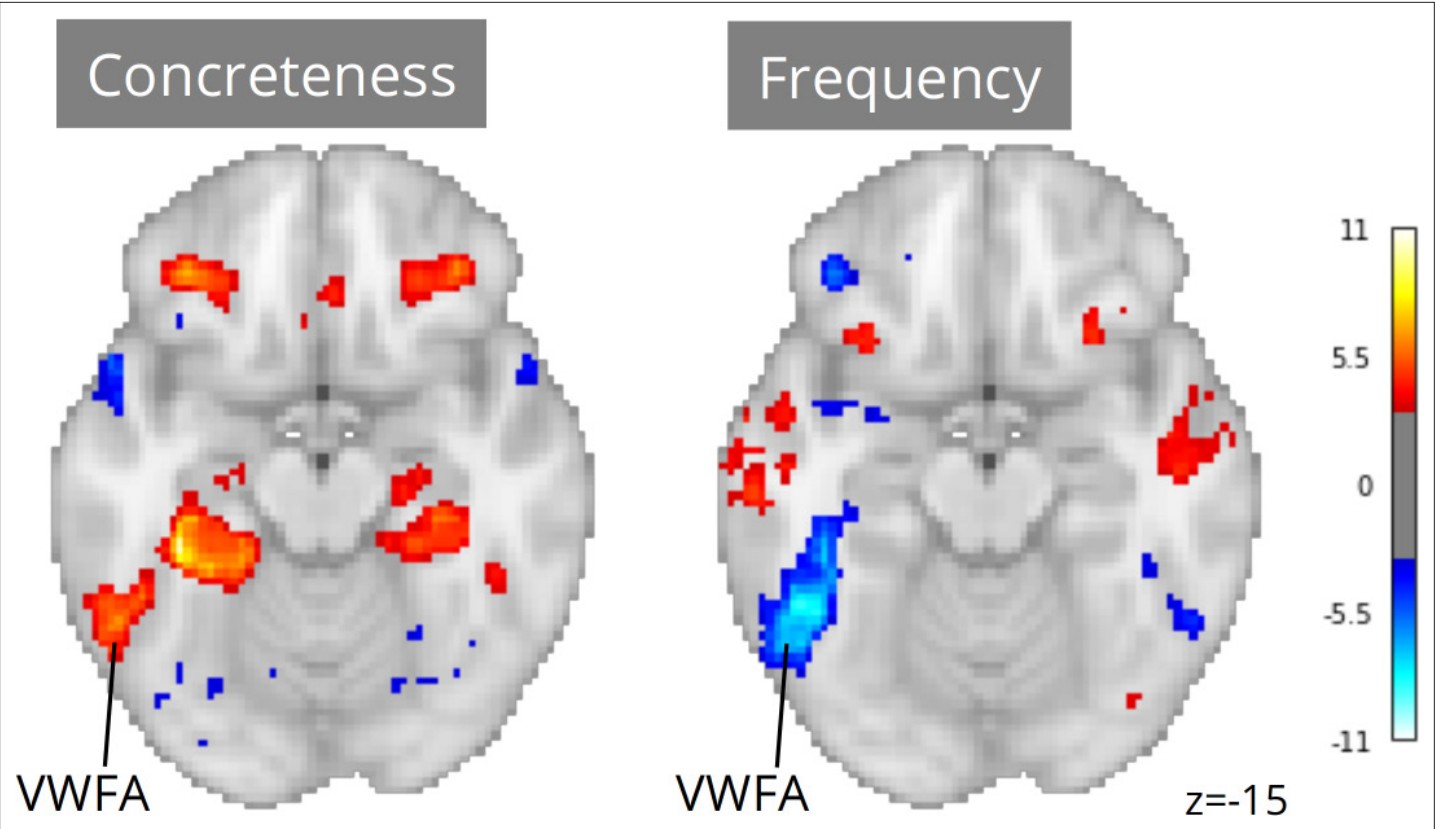

**Figure 6.** Meta-analytic statistical maps for concreteness and frequency controlling for speech, text length, number of syllables and phonemes, and phone-level Levenshtein distance. N=33 tasks; images were thresholded at Z=3.29 (p<0.001) voxel-wise. Visual word form area, VWFA.

orthographic and phonological reading subprocesses, but also modulated by modality-independent lexical-semantic properties of linguistic input. Yet, as the experiment only involved visual presentation of linguistic stimuli, this hypothesis could not be corroborated empirically. In addition, the authors observed that frequency effects disappeared when controlling for lexical concreteness. As the two variables were highly correlated, the authors speculated that the study may have lacked the power to disentangle their contributions and declared the results inconclusive.

Neuroscout makes it possible to re-evaluate linguistic hypotheses in ecological stimuli using a wide range of linguistic annotations spanning both phonological/orthographic word properties (e.g. duration and phonological distinctiveness), semantic descriptors (e.g. valence, concreteness, sensorimotor attributes), and higher-level information-theoretic properties of language sequences (e.g. entropy in next-word prediction and word-by-word surprisal). We reimplemented analytic models from *Yarkoni et al., 2008* across all Neuroscout datasets, including regressors for word frequency, concreteness, speech, and control orthographic measures (number of syllables, number of phones, and duration), alongside a standard set of nuisance parameters. As before, we used IBMA to compute meta-analytic estimates for each variable. The resulting maps displayed significant VWFA effects for both frequency and concreteness (*Figure 6*), corroborating the hypothesis of its involvement in lexical processing independent of presentation modality, and arguably in the context of language-to-imagery mapping.

Note that had we only had access to results from the original study, our conclusions might have been substantially different. Using a relatively liberal threshold of p<0.01, only 12 out of 33 tasks showed significant ROI-level association between VWFA and frequency, and only 5 tasks showed an association between VWFA and concreteness. In addition, in only one task was VWFA significantly associated with both frequency and concreteness. These ROI-level results highlight the power of scalability in the context of naturalistic fMRI analysis. By drawing on larger participant samples and more diverse stimuli, meta-analysis overcomes power and stimulus variability limitations that can cause instability in dataset-level results.

## Discussion

Neuroscout seeks to promote the widespread adoption of reproducible and generalizable fMRI research practices, allowing users to easily test a wide range of hypotheses in dozens of open naturalistic datasets using automatically extracted neural predictors. The platform is designed with a strong focus on reproducibility, providing a unified framework for fMRI analysis that reduces the burden of reproducible fMRI analysis and facilitates transparent dissemination of models and statistical results. Owing to its high degree of automation, Neuroscout also facilitates the use of meta-analytic workflows, enabling researchers to test the robustness and generalizability of their models across multiple datasets.

We have demonstrated how Neuroscout can incentivize more ecologically generalizable fMRI research by addressing common modeling challenges that have traditionally deterred naturalistic research. In particular, as we show in our meta-analyses, automatically extracted predictors can be used to test a wide range of hypotheses on naturalistic datasets without the need for costly manual annotation. Although we primarily focused on replicating established effects for validation, a range of predictors operationalizing less explored cognitive variables are already available in the platform, and, as machine learning algorithms continue to advance, we expect possibilities for interesting additions to Neuroscout's feature set to keep growing at a fast pace. As a result, we have designed Neuroscout and its underlying feature extraction framework *pliers* to facilitate community-led expansion to novel extractors— made possible by the rapid increase in public repositories of pre-trained deep learning models such as HuggingFace (*Abadi et al., 2015*) and TensorFlow Hub (*Abadi et al., 2015*).

We have also shown how Neuroscout's scalability facilitates the use of meta-analytic workflows, which enable more robust and generalizable inference. As we have pointed out in some of our examples, small participant samples and stimulus-specific effects can at times lead to misleading dataset-level results. Automatically extracted predictors are particularly powerful when paired with Neuroscout's flexible model specification and execution workflow, as their combination makes it easy to operationalize hypotheses in identical ways across multiple diverse dataset and gather more generalizable consensus estimates. While large-N studies are becoming increasingly common in cognitive neuroscience, the importance of relying on large and diverse stimulus sets has been thus far underestimated (*Westfall et al., 2016*), placing Neuroscout in a unique position in the current research landscape. Importantly, although we have primarily focused on demonstrating the advantages of large-scale workflows in the context of meta-analysis, scalability can also be leveraged for other secondary workflows (e.g. machine learning pipelines, multi-verse analyses, or mega-analyses) and along dimensions other than datasets (e.g. model parameters such as transformations and covariates).

A fundamental goal of Neuroscout is to provide researchers with tools that automatically ensure the adoption of gold-standard research practices throughout the analysis lifecycle. We have paid close attention to ensuring transparency and reproducibility of statistical modeling by adopting a community-developed specification of statistical models, and developing accessible tools to specify, visualize and execute analyses. Neuroscout's model builder can be readily accessed online, and the execution engine is designed to be portable, ensuring seamless deployment across computational environments. This is a key contribution to cognitive neuroscience, which too often falls short of meeting these basic criteria of sound scientific research.

### Challenges and future directions

A major challenge in the analysis of naturalistic stimuli is the high degree of collinearity between features, as the interpretation of individual features is dependent on co-occurring features. In many cases, controlling for confounding variables is critical for the interpretation of the primary feature— as is evident in our investigation of the relationship between FFA and face perception. However, it can also be argued that in dynamic narrative driven media (i.e. films and movies), the so-called confounds themselves encode information of interest that cannot or should not be cleanly regressed out (*Grall and Finn, 2020*).

Absent a consensus on how to model naturalistic data, we designed Neuroscout to be agnostic to the goals of the user and empower them to construct sensibly designed models through comprehensive model reports. An ongoing goal of the platform—especially as the number of features continues to increase—will be to expand the visualizations and quality control reports to enable users to better understand the predictors and their relationship. For instance, we are developing an interactive

visualization of the covariance between all features in Neuroscout that may help users discover relationships between a predictor of interest and potential confounds.

However, as the number of features continues to grow, a critical future direction for Neuroscout will be to implement statistical models which are optimized to estimate a large number of covarying targets. Of note are regularized encoding models, such as the banded-ridge regression as implemented by the Himalaya package (*Latour et al., 2022*). These models have the additional advantage of implementing feature-space selection and variance partitioning methods, which can deal with the difficult problem of model selection in highly complex feature spaces such as naturalistic stimuli. Such models are particularly useful for modeling high-dimensional embeddings, such as those produced by deep learning models. Many such extractors are already implemented in pliers and we have begun to extract and analyze these data in a prototype workflow that will soon be made widely available.

Although we have primarily focused on naturalistic datasets—as they intrinsically feature a high degree of reusability and ecological validity—Neuroscout workflows are applicable to any BIDS-compliant dataset due to the flexibility of the BIDS Stats Model specification. Indexing non-naturalistic fMRI datasets will be an important next step, an effort that will be supported by the proliferation of data sharing portals and require the widespread sharing of harmonized preprocessed derivatives that can be automatically ingested. Other important expansions include facilitating analysis execution by directly integrating with cloud-based neuroscience analysis platforms, such as https://brainlife.io/about/ (*Avesani et al., 2019*), and facilitating the collection of higher-level stimulus features by integrating with crowdsourcing platforms such as MechanicalTurk or Prolific.

In addition, as Neuroscout grows to facilitate the re-analysis of a broader set of public datasets, it will be important to reckon with the threat of 'dataset decay' which can occur from repeated sequential re-analysis (*Thompson et al., 2020*). By encouraging the central registration of all analysis attempts and the associated results, Neuroscout is designed to minimize undocumented researcher degrees of freedom and link the final published results with all previous attempts. By encouraging the public sharing of all results, we hope to encourage meta-scientists to empirically investigate statistical solutions to the problem of dataset decay, and develop methods to minimize the effect of false positives.

## Long-term sustainability

An on-going challenge for scientific software tools—especially those that rely on centralized services—is long-term maintenance, development, and user support. On-going upkeep of core tools and development of new features require a non-trivial amount of developer time. This problem is exacerbated for projects primarily supported by government funding, which generally prefers novel research to the on-going maintenance of existing tools. This is particularly challenging for centralized services, such as the Neuroscout server and web application, which require greater maintenance and coordination for upkeep.

With this in mind, we have designed many of the core components of Neuroscout with modularity as a guiding principle in order to maximize the longevity and impact of the platform. Although components of the platform are tightly integrated, they are also designed to be independently useful, increasing their general utility, and encouraging broader adoption by the community. For example, our feature extraction library (pliers) is designed for general purpose use on multimodal stimuli, and can be easily expanded to adopt novel extractors. On the analysis execution side, rather than implementing a bespoke analysis workflow, we worked to develop a general specification for statistical models under the BIDS standard (https://bids-standard.github.io/stats-models/) and a compatible execution workflow (FitLins; https://github.com/poldracklab/fitlins; *Markiewicz, 2022*). By distributing the technical debt of Neuroscout across various independently used and supported projects, we hope to maximize the robustness and impact of the platform. To ensure the community's needs are met, users are encouraged to vote on the prioritization of features by voting on issues on Neuroscout's GitHub repository, and code from new contributors is actively encouraged.

## User support and feedback

A comprehensive overview of the platform and guides for getting started can be found in the integrated Neuroscout documentation (https://neuroscout.org/docs), as well as in each tool's version-specific automatically generated documentation (hosted by ReadTheDocs, a community-supported

**Table 1.** Neuroscout datasets included in the validation analyses.

Subj is the number of unique subjects. Scan Time is the mean scan time per subject (in minutes). AV = Audio-Visual; AN = Audio Narrative.

| Name | Subj | DOI/URI | Scan time | Modality | Description |
|---|---|---|---|---|---|
| Study Forrest (*Hanke et al., 2014*) | 13 | 10.18112/openneuro. ds000113.v1.3.0 | 120 | AV | Slightly abridged German version of the movie: 'Forrest Gump' |
| Life (*Nastase et al., 2018*) | 19 | datasets.datalad. org/?dir=/labs/haxby/life | 62.8 | AV | Four segments of the Life nature documentary |
| Raiders (*Haxby et al., 2011*) | 11 | datasets.datalad. org/?dir=/labs/haxby/ raiders | 113.3 | AV | Full movie: 'Raiders of the Lost Ark' |
| Learning Temporal Structure (LTS) (*Aly et al., 2018*) | 30 | 10.18112/openneuro. ds001545.v1.1.1 | 20.1 | AV | Three clips from the movie 'Grand Budapest Hotel', presented six times each. Some clips were scrambled. |
| Sherlock (*Chen et al., 2017*) | 16 | 10.18112/openneuro. ds001132.v1.0.0 | 23.7 | AV | The first half of the first episode from 'Sherlock' TV series. |
| SherlockMerlin (*Zadbood et al., 2017*) | 18 | Temporarily unavailable | 25.1 | AV | Full episode from 'Merlin' TV series. Only used Merlin task to avoid analyzing the Sherlock task twice. |
| Schematic Narrative (*Baldassano et al., 2018*) | 31 | 10.18112/openneuro. ds001510.v2.0.2 | 50.4 | AV/AN | 16 three-minute clips, including audiovisual clips and narration. |
| ParanoiaStory (*Finn et al., 2018*) | 22 | 10.18112/openneuro. ds001338.v1.0.0 | 21.8 | AN | Audio narrative designed to elicit individual variation in suspicion/paranoia. |
| Budapest (*Visconti et al., 2020*) | 25 | 10.18112/openneuro. ds003017.v1.0.3 | 50.9 | AV | The majority of the movie 'Grand Budapest Hotel', presented in intact order |
| Naturalistic Neuroimaging Database (NNDb) (*Aliko et al., 2020*) | 86 | 10.18112/openneuro. ds002837.v2.0.0 | 112.03 | AV | Movie watching of 10 full-length movies |
| Narratives (*Nastase et al., 2021*) | 328 | 10.18112/openneuro. ds002345.v1.1.4 | 32.5 | AN | Passive listening of 16 audio narratives (two tasks were not analyzed due to preprocessing error) |

documentation platform). We plan to grow the collection of complete tutorials replicating exemplary analyses and host them in the centralized Neuroscout documentation.

Users can ask questions to developers and the community using the topic 'neuroscout' Neurostars. org— a public forum for neuroscience researchers and neuroinformatics infrastructure maintainers supported by the International Neuroinformatics Coordinating Facility (INCF). In addition, users can provide direct feedback through a form found on all pages in the Neuroscout website, which directly alerts developers to user concerns. A quarterly mailing list is also available to stay up to date with the latest feature developments in the platform. Finally, the Neuroscout developer team frequently participates at major neuroinformatics hackathons (such as Brainhack events and at major neuroimaging conferences), and plans on hosting ongoing Neuroscout-specific hackathons.

## Materials and methods
### Code availability

All code from our processing pipeline and core Neuroscout infrastructure is available online (https://www.github.com/neuroscout/neuroscout; *Alejandro de la, 2022a*), including the Python client library pyNS (https://www.github.com/neuroscout/pyNS; *Alejandro de la, 2022b*). The Neuroscout-CLI analysis engine is available as a Docker and Singularity container, and the source code is also made available (https://github.com/neuroscout/neuroscout-cli/; *Alejandro de la, 2022c*). Finally, an online supplement following the analyses showcased in this paper is available as interactive Jupyter Book (https://neuroscout.github.io/neuroscout-paper/). All are available under a permissive BSD license.

## Datasets

The analyses presented in this paper are based on 13 naturalistic fMRI datasets sourced from various open data repositories (see *Table 1*). We focused on BIDS-compliant datasets which included the exact stimuli presented with precise timing information. Datasets were queried and parsed using *pybids* (https://github.com/bids-standard/pybids; *Yarkoni et al., 2019b*; *Yarkoni et al., 2019a*) and ingested into a SQL database for further subsequent analysis. Several datasets spanned various original studies or distinct simuli (e.g. Narratives, NNDb), resulting in 35 unique 'tasks' or 'studies' available for analysis. The full list of datasets and their available predictors are available on Neuroscout (https://neuroscout.org/datasets).

## fMRI Preprocessing

Neuroscout datasets are uniformly preprocessed using FMRIPREP (version 1.2.2) (*Esteban et al., 2020*; *Esteban et al., 2019*; *Esteban et al., 2022*), a robust NiPype-based MRI preprocessing pipeline. The resulting preprocessed data are publicly available for download (https://github.com/neuroscout-datasets). The following methods description was semi-automatically generated by FMRIPREP.

Each T1-weighted (T1w) volume is corrected for intensity non-uniformity using N4BiasFieldCorrection v2.1.0 (*Tustison et al., 2010*) and skull-stripped using antsBrainExtraction.sh v2.1.0 (using the OASIS template). Spatial normalization to the ICBM 152 Nonlinear Asymmetrical template version 2009c (*Fonov et al., 2009*) is performed through nonlinear registration with the antsRegistration tool of ANTs v2.1.0 (*Avants et al., 2008*), using brain-extracted versions of both T1w volume and template. Brain tissue segmentation of cerebrospinal fluid (CSF), white matter (WM), and gray matter (GM) were performed on the brain-extracted T1w using fast (*Zhang et al., 2001*) (FSL v5.0.9).

Functional data are motion-corrected using mcflirt (FSL v5.0.9, *Jenkinson et al., 2002*). The images are subsequently co-registered to the T1w volume using boundary-based registration (*Greve and Fischl, 2009*) with 9 degrees of freedom, using flirt (FSL). Motion correcting transformations, BOLD-to-T1w transformation, and T1w-to-template warp were concatenated and applied in a single step using antsApplyTransforms (ANTs v2.1.0) using Lanczos interpolation.

Anatomically based physiological noise regressors were created using CompCor (*Behzadi et al., 2007*). A mask to exclude signals with cortical origin is obtained by eroding the brain mask, ensuring it only contains subcortical structures. Six principal components are calculated within the intersection of the subcortical mask and the union of CSF and WM masks calculated in T1w space, after their

**Table 2.** Extractor name, feature name, and description for all Neuroscout features used in the validation analyses.

| Extractor | Feature | Description |
|---|---|---|
| Brightness | brightness | Average luminosity across all pixels in each video frame. |
| Clarifai | building, landscape, text, tool | Indicators of the probability that an object belonging to each of these categories is present in the video frame. |
| FaceNet | any_faces, log_mean_time_cum | For each video frame, any_faces indicates the probability that the image displays at least one face. log_mean_time_cum indicates the cumulative time (in seconds) a given face has been on screen up since the beginning of the movie. If multiple faces are present, their cumulative time on screen is averaged. |
| Google Video Intelligence | shot_change | Binary indicator coding for shot changes. |
| FAVE/Rev | speech | Binary indicator coding for the presence of speech in the audio signal, inferred from word onsets/offsets information from force-aligned speech transcripts. |
| RMS | rms | Root mean square (RMS) energy of the audio signal. |
| Lexical norms | Log10WF, concreteness, phonlev, numsylls, numphones, duration, text_length | Logarithm of SubtlexUS lexical frequency, concreteness rating, phonological Levenshtein distance, number of syllables, number of phones, average auditory duration and number of characters for each word in the speech transcript. These metrics are extracted from lexical databases available through pliers. |

projection to the native space of each functional run. Many internal operations of FMRIPREP use Nilearn (*Abraham et al., 2014*), principally within the BOLD-processing workflow.

## Automatically extracted features

### Overview

Neuroscout leverages state-of-the-art machine learning algorithms to automatically extract hundreds of novel neural predictors from the original experimental stimuli. Automated feature extraction relies on *pliers*, a python library for multimodal feature extraction which provides a standardized interface to a diverse set of machine learning algorithms and APIs (*McNamara et al., 2017*). Feature values are ingested directly with no in place modifications, with the exception of down sampling of highly dense variables to 3 hz to facilitate storage on the server. For all analyses reported in this paper the same set of feature extractors are applied across all datasets (see *Table 2*), except where not possible due to modality mismatch (e.g. visual features in audio narratives), or features intrinsically absent from the stimuli (e.g. faces in the *Life* nature documentary). A description of all features included in this paper is provided below. A complete list of available predictors and features is actualized online at: https://neuroscout.org/predictors.

### Visual features

#### Brightness

We computed brightness (average luminosity) for frame samples of videos by computing the average luminosity for pixels across the entire image. We took the maximum value at each pixel from the RGB channels, computed the mean, and divided by 255.0 (the maximum value in RGB space), resulting in a scalar ranging from 0 to 1. This extractor is available through pliers as *BrightnessExtractor*.

#### Clarifai object detection

Clarifai is a computer vision company that specializes in using deep learning networks to annotate images through their API as a service. We used Clarifai's 'General' model, a pre-trained deep convolutional neural network (CNN) for multi-class classification of over 11,000 categories of visual concepts, including objects and themes.

To reduce the space of possible concepts, we pre-selected four concepts that could plausibly capture psychologically relevant categories (see *Table 2*). Feature extraction was performed using *pliers'* *ClarifaiAPIImageExtractor*, which wraps Clarifai's Python API client. We submitted the sampled visual frames from video stimuli to the Clarifai API, and received values representing the model's predicted probability of each concept for that frame.

#### Face detection, alignment, and recognition

Face detection, alignment, and recognition were performed using the *FaceNet* package (https://github.com/davidsandberg/facenet; *Sandberg, 2018*), which is an open TensorFlow implementation of state-of-the-art face recognition CNNs. As this feature was not natively available in *pliers*, we computed it offline and uploaded it to Neuroscout using the feature upload portal.

First, face detection, alignment, and cropping are performed through Multi-task Cascaded Convolutional Networks (MTCNN; *Zhang et al., 2016*). This framework uses unified cascaded CNNs to detect, landmark, and crop the position of a face in an image. We input sampled frames from video stimuli, and the network identified, separated, and cropped individual faces for further processing. At this step, we were able to identify if a given frame in a video contained one or more faces ('any_faces').

Next, cropped faces were input to the *FaceNet* network for facial recognition. *FaceNet* is a face recognition deep CNN based on the Inception ResNet v1 architecture that achieved state-of-the-art performance when released (*Schroff et al., 2015*). The particular recognition model we used was pre-trained on the VGGFace2 dataset (*Cao et al., 2018*), which is composed of over three million faces 'in the wild', encompassing a wide range of poses, emotions, lighting, and occlusion conditions. *FaceNet* creates a 512-dimensional embedding vector from cropped faces that represents extracted face features; thus more similar faces are closer in the euclidean embedding space.

For each dataset separately, we clustered all detected faces' embedding vectors to group together faces corresponding to distinct characters in the audio-visual videos. We used the Chinese Whispers clustering algorithm, as this algorithm subjectively grouped faces into coherent clusters better than

other commonly used algorithms (e.g. k-means clustering). Depending on the dataset, this resulted in 50–200 clusters that subjectively corresponded to readily identifiable characters across the video stimulus. For each dataset, we removed the worst-performing cluster (as for all datasets there was always one with a highly noisy profile) and grouped demonstrably different faces into one cluster. Using the generated face clusters for each dataset, we computed the cumulative time each character had been seen across the stimulus (i.e. entire movie) and log transformed the variable in order to represent the adaptation to specific faces over time. As more than one face could be shown simultaneously, we took the mean for all faces on screen in a given frame.

### Google Video Intelligence

We used the Google Video Intelligence API to identify shot changes in video stimuli. Using the *GoogleVideoAPIShotDetectionExtractor* extractor in *pliers*, we queried the API with complete video clips (typically one video per run). The algorithm separates distinct video segments, by detecting abstract shot changes in the video (i.e. the frames before and after that frame are visually different). The time at which there was a transition between two segments was given a value of 1, while all other time points received a value of 0.

### Auditory features
#### RMS

We used *librosa* (**McFee et al., 2015**), a python package for music and audio analysis, to compute root-mean-squared (RMS) as a measure of the instantaneous audio power over time, or 'loudness'.

### Speech forced alignment

For most datasets, transcripts of the speech with low-resolution or no timing information were available either from the original researcher or via closed captions in the case of commercially produced media. We force aligned the transcripts to extract word-level speech timing, using the Forced Alignment and Vowel Extraction toolkit (FAVE; **Rosenfelder et al., 2014**). FAVE employs Gaussian mixture model based monophone Hidden Markov Models (HMMs) from the Penn Phonetics Lab Forced Aligner for English (p2fa; **Yuan and Liberman, 2008**), which is based on the Hidden Markov Toolkit (**Young, 1994**). The transcripts are mapped to phone sequences with pre-trained HMM acoustic models. Frames of the audio recording are then mapped onto the acoustic models, to determine the most likely sequence. The alignment is constrained by the partial timing information available in closed captions, and the sequence present in the original transcripts. Iterative alignment continues until models converge. Linguistic features are available for all datasets except *studyforrest*, as the movie was presented in German. Transcription and annotation of stimuli in languages other than English are pending.

### Rev.com

For datasets that had no available transcript (*LearningTemporalStructure*, *SchematicNarrative*), we used a professional speech-to-text service (Rev.com) to obtain precise transcripts with word-level timing information. Rev.com provides human-created transcripts which are then force-aligned using proprietary methods to produce a high-quality, aligned transcript, similar to that generated by the FAVE algorithm.

### Speech indicator

In both cases, we binarized the resulting aligned transcripts based on word onset/offset information to produce a fine-grained speech presence feature ('speech'). These aligned transcripts served as the input to all subsequent speech-based analyses.

### Language features
#### Word frequency

Neuroscout includes a variety of frequency norms extracted from different lexical databases. For all the analyses reported here, we used frequency norms from SUBTLEX-US (**Brysbaert and New, 2009**), a 51 million words corpus of American English subtitles. The variable used in the analyses (Log10WF, see **Brysbaert and New, 2009**) is the base 10 logarithm of the number of occurrences of the word

in the corpus. In all analyses, this variable was demeaned and rescaled prior to HRF convolution. For a small percentage of words not found in the dictionary, a value of zero was applied after rescaling, effectively imputing the value as the mean word frequency. This feature was extracted using the *subtlexusfrequency dictionary* and the *PredefinedDictionaryExtractor* available in *pliers*.

### Concreteness

Concreteness norms were extracted from the (***Brysbaert et al., 2014***) concreteness database, which contains norms for over 40,000 English words, obtained from participants' ratings on a five-point scale. In all analyses, this variable was demeaned and rescaled before HRF convolution. This feature was extracted using the *concreteness* dictionary and the *PredefinedDictionaryExtractor* available in *pliers*.

### Massive auditory lexical decision norms

The Massive Auditory Lexical Decision (MALD) database (***Tucker et al., 2019***) is a large-scale auditory and production dataset that includes a variety of lexical, orthographic, and phonological descriptors for over 35,000 English words and pseudowords. MALD norms are available in Neuroscout for all words in stimulus transcripts. The analyses reported in this paper make use of the following variables:

- *Duration*: duration of spoken word in milliseconds;
- *NumPhones*: number of phones, that is of distinct speech sounds;
- *NumSylls*: number of syllables;
- *PhonLev*: mean phone-level Levenshtein distance of the spoken word from all items in the reference pronunciation dictionary, i.e. the CMU pronouncing dictionary with a few additions. This variable quantifies average phonetic similarity with the rest of the lexicon so as to account for neighborhood density and lexical competition effects (***Yarkoni et al., 2008***).

In all analyses, these variables were demeaned and rescaled before HRF convolution. MALD metrics was extracted using the *massiveauditorylexicaldecision* dictionary and the *PredefinedDictionaryExtractor* available in pliers.

### Text length

This variable corresponds to the number of characters in a word's transcription. A *TextLengthExtractor* is available in *pliers*.

## GLM models

Neuroscout uses FitLins, a newly developed workflow for executing multi-level fMRI general linear model (GLM) analyses defined by the BIDS StatsModels specification. FitLins uses *pybids* to generate run-level design matrices, and *NiPype* to encapsulate a multi-level GLM workflow. Model estimation at the first level was performed using *AFNI*—in part due to its memory efficiency—and subject and group level summary statistics were fit using the nilearn.glm module.

For all models, we included a standard set of confounds from *fmriprep*, in addition to the listed features of interest. This set includes 6 rigid-body motion-correction parameters, 6 noise components calculated using CompCor, a cosine drift model, and non-steady state volume detection, if present for that run. Using *pybids*, we convolved the regressors with an implementation of the SPM dispersion derivative haemodynamic response model, and computed first-level design matrices downsampled to the TR. We fit the design matrices to the unsmoothed registered images using a standard AR(1) + noise model.

Smoothing was applied to the resulting parameter estimate images using a 4 mm FWHM isotropic kernel. For the datasets that had more than one run per subject, we then fit a subject-level fixed-effects model with the smoothed run-level parameter estimates as inputs, resulting in subject-level parameter estimates for each regressor. Finally, we fit a group-level fixed-effects model using the previous level's parameter estimates and performed a one-sample t-test contrast for each regressor in the model.

## Meta-analysis

NiMARE (version 0.0.11rc1; available at: https://github.com/neurostuff/NiMARE; RRID:SCR_017398) was used to perform meta-analyses across the neuroscout datasets. Typical study harmonization steps

(smoothing, design matrix scaling, spatial normalization) were forgone because all group level beta and variance maps were generated using the same GLM pipeline. All group level beta and variance maps were resampled to a 2x2 × 2 mm ICBM 152 Nonlinear Symmetrical gray matter template (downloaded using *nilearn*, version 0.8.0) with linear interpolation. Resampled values were clipped to the minimum and maximum statistical values observed in the original maps. We used the DerSimonian & Laird random effects meta-regression algorithm (*DerSimonian and Laird, 1986*; *Kosmidis et al., 2017*).

## Acknowledgements

Neuroscout is made possible by funding from the National Institute of Mental Health (NIMH) of the National Institute of Health (NIH) under award number R01MH109682. In addition, research in this preprint and critical infrastructure was supported by NIMH awards P41EB019936, R24MH117179 and R01MH096906. P.H. was supported in part by the Canada First Research Excellence Fund; the Brain Canada Foundation; and Unifying Neuroscience and Artificial Intelligence - Québec.

## Additional information

### Funding

| Funder | Grant reference number | Author |
|---|---|---|
| National Institute of Mental Health | R01MH109682 | Alejandro de la Vega |
| National Institute of Mental Health | R01MH096906 | Alejandro de la Vega |
| National Institute of Mental Health | R24MH117179 | Peer Herholz Satrajit S Ghosh Ross W Blair Christopher J Markiewicz Russell A Poldrack |
| Canada First Research Excellence Fund | | Peer Herholz |
| Brain Canada Foundation | | Peer Herholz |
| Unifying Neuroscience and Artificial Intelligence | | Peer Herholz |

The funders had no role in study design, data collection and interpretation, or the decision to submit the work for publication.

### Author contributions

Alejandro de la Vega, Conceptualization, Resources, Data curation, Software, Formal analysis, Validation, Investigation, Visualization, Methodology, Writing – original draft, Project administration, Writing – review and editing; Roberta Rocca, Software, Formal analysis, Validation, Investigation, Visualization, Methodology, Writing – original draft, Writing – review and editing; Ross W Blair, Data curation, Software, Writing – review and editing; Christopher J Markiewicz, Resources, Software, Writing – review and editing; Jeff Mentch, Validation, Visualization, Writing – review and editing; James D Kent, Software, Formal analysis, Writing – review and editing; Peer Herholz, Software, Writing – review and editing; Satrajit S Ghosh, Conceptualization, Supervision, Funding acquisition, Writing – review and editing; Russell A Poldrack, Conceptualization, Funding acquisition, Writing – review and editing; Tal Yarkoni, Conceptualization, Software, Supervision, Funding acquisition, Validation, Methodology, Writing – original draft, Project administration, Writing – review and editing

### Author ORCIDs

Alejandro de la Vega ⓘ http://orcid.org/0000-0001-9062-3778
Christopher J Markiewicz ⓘ http://orcid.org/0000-0002-6533-164X
Peer Herholz ⓘ http://orcid.org/0000-0002-9840-6257

Satrajit S Ghosh http://orcid.org/0000-0002-5312-6729

**Decision letter and Author response**
Decision letter https://doi.org/10.7554/eLife.79277.sa1
Author response https://doi.org/10.7554/eLife.79277.sa2

## Additional files

### Supplementary files
• MDAR checklist

### Data availability
All code from our processing pipeline and core infrastructure is available online (https://www.github.com/neuroscout/neuroscout, copy archived at swh:1:rev:ed79e9cf4b1ee1320a2d-43c72e95f3fd3619c9b7). An online supplement including all analysis code and resulting images is available as a public GitHub repository (https://github.com/neuroscout/neuroscout-paper, copy archived at swh:1:rev:abd64777a96bdcefdfdf40edc70ff6ed937a5bcc). All analysis results are made publicly available in a public GitHub repository (https://github.com/neuroscout/neuroscout-paper).

The following previously published datasets were used:

| Author(s) | Year | Dataset title | Dataset URL | Database and Identifier |
|---|---|---|---|---|
| Hanke M | 2014 | studyforrest | https://doi.org/10.18112/openneuro.ds000113.v1.3.0 | OpenNeuro, 10.18112/openneuro.ds000113.v1.3.0 |
| Nastase SA | 2017 | Neural responses to naturalistic clips of animals | https://datasets.datalad.org/?dir=/labs/haxby/life | DataLad, /labs/haxby/life |
| Haxby JV | 2017 | Dartmouth Raiders Dataset | https://datasets.datalad.org/?dir=/labs/haxby/raiders | DataLad, /labs/haxby/raiders |
| Aly M, Chen J, Turk-Browne NB, Hasson U | 2018 | Learning Temporal Structure | https://doi.org/10.18112/openneuro.ds001545.v1.1.1 | OpenNeuro, 10.18112/openneuro.ds001545.v1.1.1 |
| Chen J, Leong YC, Honey CJ, Yong CH, Norman KA, Hasson U | 2017 | Sherlock | https://doi.org/10.18112/openneuro.ds001132.v1.0.0 | OpenNeuro, 10.18112/openneuro.ds001132.v1.0.0 |
| Zadbood A, Chen J, Leong YC, Norman KA, Hasson U | 2017 | SherlockMerlin | https://openneuro.org/datasets/ds001110/versions/00003 | OpenNeuro, 10.18112/openneuro.ds001110 |
| Baldassano C, Hasson U, Norman KA | 2018 | Schematic Narrative | https://doi.org/10.18112/openneuro.ds001510.v2.0.2 | OpenNeuro, 10.18112/openneuro.ds001510.v2.0.2 |
| Finn ES, Corlett PR, Chen G, Bandettini PA, Constable RT | 2018 | ParanoiaStory | https://doi.org/10.18112/openneuro.ds001338.v1.0.0 | OpenNeuro, 10.18112/openneuro.ds001338.v1.0.0 |
| Visconti di Oleggio Castello M, Chauhan V, Jiahui G, Gobbini MI | 2020 | Budapest | https://doi.org/10.18112/openneuro.ds003017.v1.0.3 | OpenNeuro, 10.18112/openneuro.ds003017.v1.0.3 |
| Aliko S, Huang J, Gheorghiu F, Meliss S, Skipper JI | 2020 | Naturalistic Neuroimaging Database | https://doi.org/10.18112/openneuro.ds002837.v2.0.0 | OpenNeuro, 10.18112/openneuro.ds002837.v2.0.0 |
| Nastase SA | 2021 | Narratives | https://doi.org/10.18112/openneuro.ds002345.v1.1.4 | OpenNeuro, 10.18112/openneuro.ds002345.v1.1.4 |

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
