## [Editor Report]

This is an important, methodologically compelling paper. It describes a powerful new online software platform for analysing data from naturalistic fMRI studies. The paper describes both the philosophy behind and intended usage of the software, and offers several examples of the types of results that can be computed using publicly available datasets. It will provide an important new tool for the open neuroscience community who are seeking to perform standardised and reproducible analyses of naturalistic fMRI datasets.

---

## [Decision Letter]

**Decision letter after peer review:**

Thank you for submitting your article "Neuroscout, a unified platform for generalizable and reproducible fMRI research" for consideration by *eLife*. Your article has been reviewed by 3 peer reviewers, and the evaluation has been overseen by Laurence Hunt as the Reviewing Editor and Tamar Makin as the Senior Editor. The following individuals involved in the review of your submission have agreed to reveal their identity: Emily Finn (Reviewer #1); Christopher Baldassano (Reviewer #2); Eugene P Duff (Reviewer #3).

Essential revisions:

All reviewers were in agreement that Neuroscout will provide a valuable tool to the neuroimaging research community, that the structure of the platform is a novel one (that offers unique functionality not currently available on other Open Neuroimaging platforms), and that the manuscript is well-written. Following discussion with the reviewers we agreed that, while the authors should provide a point-by-point rebuttal to the reviewers' comments below, the three areas that should receive the highest priority in revising the manuscript are:

1) Provide slightly more in the way of assistance/help for users who are completely new to the platform to ensure that they can 'get started' without encountering errors – e.g. two of the reviewers (#2 and #3) attempted to use the platform but encountered errors when getting started, which could easily put off new users from adopting Neuroscout.

2) Discuss more extensively whether there are plans for further expansion/what the ultimate 'scope' of the project is beyond GLM specification. In particular, provide a specific discussion of how exactly the authors/developers see the software evolving over the short-, medium-, and long-term, and any plans they have in place to ensure continued development and promote a robust user community. Of course, it's hard to predict uptake with any new software platform, and this manuscript is an important step in publicizing the platform, but where do we go from here? Will there be a dedicated developer or team of developers moving forward? If so, how do they plan to prioritize and execute new features -- i.e., top-down decisions about important extensions (more detail on what these are?) versus bottom-up responses to user requests? If not, how will the authors/original developers ensure the continued health of the software? What kind of user support/promotion will exist -- passive usage monitoring? regular user surveys? active mailing/discussion lists? hackathons? etc. etc.

3) Address some of the specific technical questions about how Neuroscout handles certain aspects of the GLM, as highlighted in particular by reviewer #1.

*Reviewer #2 (Recommendations for the authors):*

The Neuroscout platform and the manuscript are very impressive, and I have no comments on how the presentation of the material could be improved.

I attempted to perform an analysis in Neuroscout as part of my review and was not successful. I was able to set up a simple analysis which successfully validated and compiled. To run the analysis, I first tried using Docker, as recommended. When doing so, I was prompted to enter my username and password for github.com. This was confusing to me since a github login was not discussed as part of this system. Entering my github credentials produced an error, since "Support for password authentication was removed on August 13, 2021. Please use a personal access token instead."

I also tried to run the analysis via singularity. The command recommended in the documentation was "singularity pull oras://ghcr.io/neuroscout/neuroscout-cli:" which yielded the error "FATAL: While pulling image from oci registry: failed to get checksum for //ghcr.io/neuroscout/neuroscout-cli:master: no layer found corresponding to SIF image". I instead built a singularity image from docker using "singularity build neuroscout.simg docker://neuroscout/neuroscout-cli" which ran successfully. To run the singularity image, the documentation incorrectly left out the second "run" argument to the command, but after adding that in and running "singularity run --cleanenv neuroscout.simg run [id] testns/" the analysis started. Unfortunately, I then hit the same github issue described above and did not know how to troubleshoot the issue.

The fact that analyses must be run locally is understandable since it greatly decreases the cost and complexity of running the Neuroscout website. However, it could increase adoption of the platform if a cloud-based execution option were available in some form – I'm unsure if there are options for easily spinning up free (or low-cost) cloud instances that could allow users without local computing resources to run analyses.

There is a sentence in the manuscript that I did not understand: "Interestingly, these effects were robust to phonological and orthographic covariates, suggesting that the involvement of VWFA in language comprehension may not be specific to reading." This sentence is referring to a study that is described as a "reading experiment", so I do not understand how the results could show that VWFA is involved in tasks besides reading.

*Reviewer #3 (Recommendations for the authors):*

The manuscript is prepared to a high standard. I have a number of recommendations:

– It would be nice to see more of a review of similar attempts at web analysis platforms and related software – possibly including those outside of imaging.

– Furthermore, general readers may appreciate a little more of a survey of the open fMRI analysis ecosystem that this is based upon, in the intro/results. Possibly further more detailed schematic figures could help.

– The description of the platform as "end-to-end" might be qualified – for me it produced unrealistic expectations for the breadth of the analyses that could be specified.

– More discussion on how this tool will be maintained and developed, including mechanisms for community contributions to its various elements.

– The example analyses were well chosen, but I sometimes lost track of what they were demonstrating.

– My attempt to access the analyses on binder seemed to fail.

– There was also a possible typo in the git command when I tried to pull a dataset ("fatal: repository 'https://github.com/neuroscout-datasets/SemanticNarrative-/' not found'").

– I think the manuscript would be strengthened with a broader discussion of future possibilities and goals for the tool. For example, more detail on the challenges of moving beyond naturalistic stimuli, and extension to modalities other than fMRI.

– Some discussion of the inferential challenges related to the re-analysis of datasets could be warranted.

---

## [Author Response]

Essential revisions:All reviewers were in agreement that Neuroscout will provide a valuable tool to the neuroimaging research community, that the structure of the platform is a novel one (that offers unique functionality not currently available on other Open Neuroimaging platforms), and that the manuscript is well-written. Following discussion with the reviewers we agreed that, while the authors should provide a point-by-point rebuttal to the reviewers' comments below, the three areas that should receive the highest priority in revising the manuscript are:1) Provide slightly more in the way of assistance/help for users who are completely new to the platform to ensure that they can 'get started' without encountering errors – e.g. two of the reviewers (#2 and #3) attempted to use the platform but encountered errors when getting started, which could easily put off new users from adopting Neuroscout.

We agree that the first-time user experience is critical to the success of any scientific software project. As such, we have launched a comprehensive documentation website for the Neuroscout project (https://neuroscout.org/docs) which integrates our three major ecosystem tools (neuroscout.org; Neuroscout-CLI, and the PyNS Python API). In addition, for Neuroscout-CLI and PyNS, we have launched versioned documentation to ensure up-to-date usage references are available in a user-friendly format

(https://pyns.readthedocs.io/en/latest/; https://neuroscout-cli.readthedocs.io/en/latest/).

In addition, to help guide new users, we have launched a fully cloud-based workflow using Google Colab— a free Jupyter notebook environment. Users can use this free resource to run analyses designed on neuroscout.org without using any of their local compute resources, or having to install any software. This option is clearly documented in the new Neuroscout documentation, and linked in the “Run” tab of the Neuroscout model builder.

We mention this new documentation in the introduction to our Results:

“Complete and up-to-date documentation of all of the platform’s components, including

Getting Started guides to facilitate first time users, is available in the official Neuroscout

Documentation (https://neuroscout.org/docs).“ pg. 3

As well as in the Discussion section in a new section on User support and feedback:

“A comprehensive overview of the platform and guides for getting started can be found in the integrated Neuroscout documentation (https://neuroscout.org/docs), as well as in each tool’s version-specific automatically generated documentation (hosted by ReadTheDocs, a community-supported documentation platform). We plan to grow the collection of complete tutorials replicating exemplary analyses and host them in the centralized Neuroscout documentation. “ (pg. 12)

Finally, we’re happy to report that by communicating with the reviewers, we were able to identify a bug which has since been corrected and should no longer be an obstacle to new users.

2) Discuss more extensively whether there are plans for further expansion/what the ultimate 'scope' of the project is beyond GLM specification. In particular, provide a specific discussion of how exactly the authors/developers see the software evolving over the short-, medium-, and long-term, and any plans they have in place to ensure continued development and promote a robust user community. Of course, it's hard to predict uptake with any new software platform, and this manuscript is an important step in publicizing the platform, but where do we go from here? Will there be a dedicated developer or team of developers moving forward? If so, how do they plan to prioritize and execute new features -- i.e., top-down decisions about important extensions (more detail on what these are?) versus bottom-up responses to user requests? If not, how will the authors/original developers ensure the continued health of the software? What kind of user support/promotion will exist -- passive usage monitoring? regular user surveys? active mailing/discussion lists? hackathons? etc. etc.

The reviewers are correct to point out that planning for short and long term extensibility and support of the project is critical to long term success. To address these concerns, we have explicitly addressed this in the Discussion section in three sections.

First, we address long term sustainability in a new Discussion section:

“An on-going challenge for scientific software tools—especially those that rely on centralized services—is long-term maintenance, development and user support. On-going upkeep of core tools and development of new features require a non-trivial amount of developer time. This problem is exacerbated for projects primarily supported by government funding, which generally prefers novel research to the on-going maintenance of existing tools. This is particularly challenging for centralized services, such as the Neuroscout server and web application, which require greater maintenance and coordination for upkeep.

With this in mind, we have designed many of the core components of Neuroscout with modularity as a guiding principle in order to maximize the longevity and impact of the platform. Although components of the platform are tightly integrated, they are also designed to be independently useful, increasing their general utility, and encouraging broader adoption by the community. For example, our feature extraction library (pliers) is designed for general purpose use on multimodal stimuli, and can be easily expanded to adopt novel extractors. On the analysis execution side, rather than implementing a bespoke analysis workflow, we worked to develop a general specification for statistical models under the BIDS standard (BIDS StatsModels; https://bids-standard.github.io/stats-models/) and a compatible execution workflow (FitLins; https://github.com/poldracklab/fitlins). By distributing the technical debt of Neuroscout across various independently used and supported projects, we hope to maximize the robustness and impact of the platform. To ensure the community’s needs are met, users are encouraged to vote on the prioritization of features by voting on issues on Neuroscout’s GitHub repository, and code from new contributors is actively encouraged.” (pg. 12)

Next, in the aforementioned User support and feedback section we more explicitly discuss resources available to users, and community involvement:

“Users can ask questions to developers and the community using the topic ‘neuroscout’ on Neurostars.org— a public forum for neuroscience researchers and neuroinformatics infrastructure maintainers managed by the INCF. In addition, users can provide direct feedback through a form found on all pages in the Neuroscout website, which directly alerts developers to user concerns. A mailing list is also available to stay up to date with the latest feature developments in the platform. Finally, the Neuroscout developer team frequently participates at major neuroinformatics hackathons (such as Brainhack events and at major neuroimaging conferences) and plans on hosting ongoing Neuroscout specific hackathons.” (pg. 12)

Finally, in the revamped Neuroscout Documentation website, we have added a section “Get involved”, which explicitly documents how to ask questions, report bugs, request features or contribute to the project (https://neuroscout.org/docs/overview/get_involved.html). We plan on keeping this section actively updated, to reflect events where the team will be present, or hackathons hosted specifically for Neuroscout.

3) Address some of the specific technical questions about how Neuroscout handles certain aspects of the GLM, as highlighted in particular by reviewer #1.

See our responses to Reviewer 1 for a detailed response to these points. We have expanded our discussion to better guide users in these difficult methodological issues.

Reviewer #2 (Recommendations for the authors):The Neuroscout platform and the manuscript are very impressive, and I have no major comments on how the presentation of the material could be improved.I attempted to perform an analysis in Neuroscout as part of my review and was not successful. I was able to set up a simple analysis which successfully validated and compiled. To run the analysis, I first tried using Docker, as recommended. When doing so, I was prompted to enter my username and password for github.com. This was confusing to me since a github login was not discussed as part of this system. Entering my github credentials produced an error, since "Support for password authentication was removed on August 13, 2021. Please use a personal access token instead."I also tried to run the analysis via singularity. The command recommended in the documentation was "singularity pull oras://ghcr.io/neuroscout/neuroscout-cli:" which yielded the error "FATAL: While pulling image from oci registry: failed to get checksum for //ghcr.io/neuroscout/neuroscout-cli:master: no layer found corresponding to SIF image". I instead built a singularity image from docker using "singularity build neuroscout.simg docker://neuroscout/neuroscout-cli" which ran successfully. To run the singularity image, the documentation incorrectly left out the second "run" argument to the command, but after adding that in and running "singularity run --cleanenv neuroscout.simg run [id] testns/" the analysis started. Unfortunately, I then hit the same github issue described above and did not know how to troubleshoot the issue.The fact that analyses must be run locally is understandable since it greatly decreases the cost and complexity of running the Neuroscout website. However, it could increase adoption of the platform if a cloud-based execution option were available in some form – I'm unsure if there are options for easily spinning up free (or low-cost) cloud instances that could allow users without local computing resources to run analyses.

We again thank the reviewer for their detailed report that allowed us to address these issues. We are also happy to report that we have developed a notebook workflow for Neuroscout which can be run for free in the Google Collab cloud. This is documented in our revamped documentation, and linked in the Run page of the Neuroscout application now: https://neuroscout.github.io/neuroscout-docs//cli/Neuroscout_CLI_Colab_Demo.html. We are also actively pursuing integration with Brainlife.io, a cloud-based neuroscience analysis platform.

We mention the future Brainlife integration in the Future Directions section:

“Other important expansions include facilitating analysis execution by directly integrating with cloud-based neuroscience analysis platforms, such as Brainlife.io (Avesani et al., 2019)…” (pp. 11)

There is a sentence in the manuscript that I did not understand: "Interestingly, these effects were robust to phonological and orthographic covariates, suggesting that the involvement of VWFA in language comprehension may not be specific to reading." This sentence is referring to a study that is described as a "reading experiment", so I do not understand how the results could show that VWFA is involved in tasks besides reading.

We thank the reviewer for this comment, and have rephrased as follows for more clarity:

“Interestingly, these effects were robust to phonological and orthographic covariates, suggesting that VWFA activity may not only be involved in orthographic and phonological reading subprocesses, but also modulated by modality-independent lexical-semantic properties of linguistic input. Yet, as the experiment only involved visual presentation of linguistic stimuli, this hypothesis could not be corroborated empirically.” (pp. 9)

Reviewer #3 (Recommendations for the authors):The manuscript is prepared to a high standard. I have a number of recommendations:– It would be nice to see more of a review of similar attempts at web analysis platforms and related software – possibly including those outside of imaging.

To our knowledge, there are scant efforts in neuroimaging to provide web-based analysis platforms. The vast majority of work has been to develop data sharing portals (such as OpenNeuro), and reproducible tools that pipelines that can be put together by users on their own computational resources. One notable exception is Brainlife.io, which we have added to the introduction. Although we agree that it might be helpful to review other similar efforts outside of neuroimaging, we feel that we are not sufficiently familiar with them to confidently introduce them here. Instead, we have chosen to review the open fMRI analysis ecosystem as suggested by the reviewer in their second suggestion.

– Furthermore, general readers may appreciate a little more of a survey of the open fMRI analysis ecosystem that this is based upon, in the intro/results. Possibly further more detailed schematic figures could help.

We agree that we did not introduce the ecosystem in sufficient detail. We added a paragraph to the introduction to better review the existing ecosystem, and position Neuroscout within it. We appreciate this suggestion, as it makes it more clear to readers exactly which part of the analysis lifecycle Neuroscout contributes to.

“The recent proliferation of community-led tools and standards—most notably the Brain Imaging Data Structure (Gorgolewski et al., 2016) standard—has galvanized efforts to foster reproducible practices across the data analysis lifecycle. A growing number of data archives, such as OpenNeuro (Markiewicz, Gorgolewski, et al., 2021), now host hundreds of publicly available neuroimaging datasets, including dozens of naturalistic fMRI datasets. The development of standardized quality control and preprocessing pipelines, such as MRIQC (Esteban et al., 2017), fmriprep (Esteban et al., 2019, 2022), and C-PAC (Craddock et al., 2013), facilitate their analysis and can be launched on emerging cloud-based platforms, such as Brainlife.io (Avesani et al., 2019). However, fMRI model specification and estimation remains challenging to standardize, and typically results in bespoke modeling pipelines that are not often shared, and can be difficult to re-use. Unfortunately, despite the availability of a rich ecosystem of tools, assembling them into a complete and reproducible workflow remains out of reach for many scientists due to substantial technical challenges.” (pp. 2)

– The description of the platform as "end-to-end" might be qualified – for me it produced unrealistic expectations for the breadth of the analyses that could be specified.

We agree with the reviewer that “end-to-end” implies that the entire analysis can be run and finished on the cloud. In order to avoid this implication, we have replaced “end-to-end” with the term “unified”, which is also consistent with our description of the platform in the title.

– More discussion on how this tool will be maintained and developed, including mechanisms for community contributions to its various elements.

Thank you for this important comment. We have added an extensive section in the

Discussion to discuss the long-term maintainability of the platform, and future directions (quote above in the response to editor). In addition, in our revamped documentation, we dedicate a page to “Getting involved” in order to orient new users and potential contributors to our development practices: https://neuroscout.org/docs/overview/get_involved.html

– The example analyses were well chosen, but I sometimes lost track of what they were demonstrating.

We appreciate this feedback, however we had a difficult time identifying a useful change to make. If the reviewer has any additional feedback here to help us refine our examples, we would be happy to make changes.

– My attempt to access the analyses on binder seemed to fail.

Unfortunately, the Binder service can fail on occasion with no repeatable cause. We have checked the JupyterBook to ensure there are no major outstanding issues that would cause this behavior. Typically, re-launching Binder will solve the problem.

– There was also a possible typo in the git command when I tried to pull a dataset ("fatal: repository 'https://github.com/neuroscout-datasets/SemanticNarrative-/' not found'").

We thank the reviewer for this issue report. This issue was also noticed by Reviewer 2, and we have since addressed it. Thankfully it only affected a subset of datasets, but we very much appreciate the in-depth testing of the platform by the reviewer.

– I think the manuscript would be strengthened with a broader discussion of future possibilities and goals for the tool. For example, more detail on the challenges of moving beyond naturalistic stimuli, and extension to modalities other than fMRI.

We have substantially expanded the discussion with a section on “Long term sustainability”, as well as a clearly defined “Challenges and future directions” section which details the upcoming priorities for Neuroscout. In particular, we chose to expand on three issues: handling collinearity and implementing more sophisticated statistical modeling workflows that are a better fit for naturalistic data.

With respect to moving beyond naturalistic stimuli, the primary challenge is related to the availability of pre-processed derivatives at scale on data sharing portals such as OpenNeuro. Neuroscout already has the ability to ingest non-naturalistic datasets, we simply chose to focus on naturalistic datasets as these datasets are inherently more flexible and potentially generalizable.

We tried to clarify that Neuroscout is compatible with this next step by revising this sentence:

“Although we have primarily focused on naturalistic datasets—as they intrinsically feature a high degree of reusability and ecological validity—Neuroscout workflows are applicable to any BIDS-compliant dataset due to the flexibility of the BIDS Stats Model specification. Indexing non-naturalistic fMRI datasets will be an important next step, an effort that will require the widespread sharing of harmonized pre-processed derivatives that can be automatically ingested.” (pp. 11)

– Some discussion of the inferential challenges related to the re-analysis of datasets could be warranted.

The reviewer brings up relevant concerns regarding long-term re-use of public datasets, and what is being called “dataset decay”. An important principle guiding Neuroscout is to minimize undocumented researcher degrees of freedom. Unlike traditional re-analysis of a public dataset, in which only the final analysis would be shared with the public, Neuroscout “locks” finalized analyses. If users want to modify their analysis, they must “clone” that analysis, keeping alive a record of their previous analysis attempts. As such, there exists a trail of all analyses that preceded the “final” analysis that is shared and published. Thus, although we have not formally implemented any correction for multiple comparisons to combat dataset decay, by keeping track of all attempted analysis, we may be able to investigate this issue in more detail in the future. In addition, we believe that by placing a focus on meta-analytic approaches, Neuroscout minimizes over-reliance on results from any one dataset.

We mention this issue as a future challenge under “Challenges and future directions”

“In addition, as Neuroscout grows to facilitate the re-analysis of a broader set of public datasets, it will be important to reckon with the threat of “dataset decay” which can occur from repeated sequential re-analysis (Thompson, Wright, Bissett, and Poldrack, 2020). By encouraging the central registration of all analysis attempts and the associated results, Neuroscout is designed to minimize undocumented researcher degrees of freedom and link the final published results with all previous attempts. By encouraging the public sharing of all results, we hope to encourage meta-scientists to empirically investigate statistical solutions to the problem of dataset decay and develop methods to minimize the effect of false positives.” (pp. xx)